# TaskMet: Task-Driven Metric Learning for Model Learning

**Dishank Bansal**[*]   **Ricky T. Q. Chen**   **Mustafa Mukadam**   **Brandon Amos**
Meta

## Abstract

Deep learning models are often used with some downstream task. Models solely trained to achieve accurate predictions may struggle to perform well on the desired downstream tasks. We propose using the task loss to learn a metric which parameterizes a loss to train the model. This approach does not alter the optimal prediction model itself, but rather changes the model learning to emphasize the information important for the downstream task. This enables us to achieve the best of both worlds: a prediction model trained in the original prediction space while also being valuable for the desired downstream task. We validate our approach through experiments conducted in two main settings: 1) decision-focused model learning scenarios involving portfolio optimization and budget allocation, and 2) reinforcement learning in noisy environments with distracting states. The source code to reproduce our experiments is available here.

## 1   Introduction

Machine learning models for prediction are typically trained to maximize the likelihood on a training dataset. While the models are capable of universally approximating the underlying data generating process to predict the output, they are prone to approximation errors due to limited training data and model capacity. These errors lead to suboptimal performance in downstream tasks where the models are used. Furthermore, even though a model may appear to have reasonable predictive performance on the metric and training data it was trained on, such as the mean squared error, employing the model for a downstream task may require the model to focus on different parts of the data that were not emphasized in the training for predictive performance. Overcoming the discrepancy between the model's prediction task and performance on a downstream task is the focus of our paper.

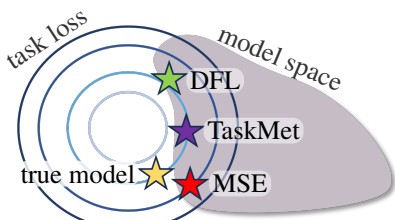

Figure 1: The *MSE* results in a model close to the true model in the prediction space, but may give poor task performance. *Decision-focused learning* (DFL) methods optimize the task loss, but may deviate from the prediction space. *TaskMet* optimizes the task loss while retaining the prediction task.

Examples of settings where the model's prediction loss $\mathcal{L}_{\text{pred}}$ is mis-matched from the downstream task $\mathcal{L}_{\text{task}}$ include the following, which table 1 also summarizes:

1. the *portfolio optimization* setting from Wilder et al. [2019], which predicts the expected returns from stocks for a financial portfolio. Here, the $\mathcal{L}_{\text{pred}}$ is the MSE and $\mathcal{L}_{\text{task}}$ is from the regret of running a portfolio optimization problem on the output;

2. the *allocation* setting from Wilder et al. [2019], which predicts the value of items that are being allocated, e.g. click-through-rates for recommender systems. Here, $\mathcal{L}_{\text{pred}}$ is the MSE and $\mathcal{L}_{\text{task}}$ measures the result of allocating the highest-value items.

---

[*]Work done as part of the Meta AI residency program.

37th Conference on Neural Information Processing Systems (NeurIPS 2023).

Table 1: Settings we focus on where there is a discrepancy between the prediction task of a model and the downstream task where the model is deployed, i.e., $\mathcal{L}_{\text{pred}} \neq \mathcal{L}_{\text{task}}$.

| Setting | $(x)$
Features | $(y)$
Predictions | $(\mathcal{L}_{\text{pred}})$
Prediction Loss | $(\mathcal{L}_{\text{task}})$
Task Loss |
|---|---|---|---|---|
| Portfolio optimization | Stock information | Expected return of a stock | MSE | Portfolio's performance |
| Budget allocation | Item information | Value of item | MSE | Allocation's performance |
| Model-based RL | Current state and action | Next state | MSE | Value estimation given the model |

3. the *model-based reinforcement learning* setting of learning the system dynamics from Nikishin et al. [2022]. Here, $\mathcal{L}_{\text{pred}}$ is the MSE of dynamics model and the $\mathcal{L}_{\text{task}}$ measures how well the agent performs for downstream value predictions.

Motivated by examples such as in table 1, the research topics of *end-to-end task-based model learning* [Bengio, 1997, Donti et al., 2017], *decision-focused learning* [Wilder et al., 2019], and *Smart "Predict, then Optimize"* [Elmachtoub and Grigas, 2022] study how to use information from the downstream task to improve the model's performance on that particular task. Task-based learning has applications in financial price predictions [Bengio, 1997, Elmachtoub and Grigas, 2022], inventory stock, demand, and price forecasting [Donti et al., 2017, Elmachtoub and Grigas, 2022, El Balghiti et al., 2019, Mandi et al., 2020, Liu et al., 2023], dynamics modeling for model-based reinforcement learning [Farahmand et al., 2017, Amos et al., 2018, Farahmand, 2018, Bhardwaj et al., 2020, Voelcker et al., 2022, Nikishin et al., 2022], renewable nowcasting [Vohra et al., 2023], vehicular routing [Shi and Tokekar, 2023], restless multi-armed bandits for maternal and child care [Wang et al., 2022], medical resource allocation [Chung et al., 2022], and budget allocation, matching, and recommendation problems [Kang et al., 2019, Wilder et al., 2019, Shah et al., 2022].

**Limitations of task-based learning.** Task-based model learning comes with the goal of being able to discover task-relevant features and data-samples on its own without the need of explicit inductive biases. The current trend for end-to-end model learning uses task loss along with the prediction loss to train the prediction models. Though easy to use, these methods may be limited by 1) the prediction overfitting to the particular task, rendering it unable to generalize; 2) the need to tuning the weight combining the task and prediction losses as in eq. (1).

**Our contributions.** We propose one way of overcoming these limitations: use the task-based learning signal not to directly optimize the weights of the model, but to *shape* a prediction loss that is constructed in a way so that the model will always stay in the original prediction space. We do this in section 3 via metric learning in the prediction space and use the task signal to learn a parameterized Mahalanobis loss. This enables more interpretable learning of the model using the metric compared to learning with a combination of task loss and prediction loss. The learned metric can uncover underlying properties of the task that are useful for training the model, e.g. as in figs. 4 and 7. Section 4 shows the empirical success of metric learning on decision focused model learning and model-based reinforcement learning. Figure 1 illustrates the differences to prior methods.

## 2 Background and related work

**Task-based model learning**. We will mostly focus on solving regression problems where the dataset $\mathcal{D} := \{(x_i, y_i)\}_{i=1}^{N}$ consists of $N$ input-output pairs, which we will assume to be in Euclidean space. The model makes a prediction $\hat{y} := f_\theta(x)$ and is parameterized by $\theta$. The model has an associated prediction loss, $\mathcal{L}_{\text{pred}}$, and is used in conjunction with some downstream task that provides a task loss, $\mathcal{L}_{\text{task}}$, which characterizes how well the model performs on the task. The most relevant related work to ours includes the approaches of Bengio [1997], Donti et al. [2017], Farahmand et al. [2017], Kang et al. [2019], Wilder et al. [2019], Nikishin et al. [2022], Shah et al. [2022], Voelcker et al. [2022], Nikishin et al. [2022], Anonymous [2023], Shah et al. [2023], which learn the optimal prediction model parameter $\theta$ to minimize the task loss $\mathcal{L}_{\text{task}}$:

$$\theta^\star := \arg\min_\theta \mathcal{L}_{\text{task}}(\theta) + \alpha \mathcal{L}_{\text{pred}}(\theta), \tag{1}$$

where $\alpha$ is a regularization parameter to weigh the prediction loss which is MSE error (eq. (2)) in general. Alternatives to eq. (1) include 1) *Smart, "Predict, then Optimize"* (SPO) methods [Elmachtoub and Grigas, 2022, El Balghiti et al., 2019, Mandi et al., 2020, Liu et al., 2023], which

consider surrogates for when the derivative is undefined or uninformative, or 2) changing the prediction space from the original domain into a latent domain with task information, e.g. task-specific latent dynamics for RL [Hafner et al., 2019b,a, Hansen et al., 2022]. Extensions such as Gupta and Zhang [2023], Zharmagambetov et al. [2023], Ferber et al. [2023] learn surrogates to overcome computationally expensive losses in eq. (1). Sadana et al. [2023] provide a further survey of this research area.

Separate from above line of work, the computer vision and NLP communities have also considered task-based losses for models: [Pinto et al., 2023] tune vision models with task rewards, e.g. for detection, segmentation, colorization, and captioning; Wu et al. [2021] consider representation learning for multiple tasks, Fernando and Tsokos [2021], Phan and Yamamoto [2020] consider weighted loss for class imbalance problems in classification, object detection.

Works such as Farahmand et al. [2017], Voelcker et al. [2022] use task loss in a different way compared to the above methods. They use task loss as a weighting term in the MSE loss itself. So the models are trained to focus more on samples with higher task loss. In their work, the task is the estimation of the value function in model-based RL. This can be seen as the instantiation of our work where the task loss is directly used as a metric instead of learning a metric.

Other related work on metric learning such as Hastie and Tibshirani [1995], Yang and Jin [2006], Weinberger and Tesauro [2007], Kulis et al. [2013], Hauberg et al. [2012], Kaya and Bilge [2019] often learns a non-Euclidean metric or distance that captures the geometry of the data and then solves a prediction task such as regression, clustering, or classification in that geometry. Other methods such as Voelcker et al. [2022] can handcraft metrics based on domain knowledge. In contrast to these, in the task-based model learning, we propose that the downstream task (instead of the data alone) gives the relevant metric for the prediction, and that it is possible to use end-to-end learning as in eq. (4) to obtain the task-based metric.

**How our contribution fits in.** The mentioned methods mainly deal with using task-based losses to condition the model learning either by differentiation through task loss to update the model or using it directly as weighing for MSE prediction loss. Whereas our work focuses on using task loss to *learn* a parameterized prediction loss which is then used to train the model. The task loss is not *directly* used for model training.

## 3    Task-driven metric learning for model learning

We first present why it's useful to see the prediction space as a non-Euclidean metric space with an unknown metric, then show how task-based learning methods can be used to learn that metric.

### 3.1    Metrics in the prediction space — Mahalanobis losses

When defining a loss on the model, we are forced to make a choice about the geometry to quantify how good a prediction is. This geometric information is often implicitly set in standard learning settings and there are often no other reasonable choices without more information. For example, a supervised model $f_\theta$ parameterized by $\theta$ is often trained with the mean squared error (MSE)

$$\theta^\star_{\text{MSE}} := \arg\min_\theta \mathbb{E}_{(x,y)\sim\mathcal{D}} \left[ (f_\theta(x) - y)^2 \right]. \tag{2}$$

The MSE makes the assumption that the geometry of the prediction space is Euclidean. While it is a natural choice, it may not be ideal when the model needs to focus on important parts of the data that are under-emphasized under the Euclidean metric. This could come up by needing to emphasize some samples over others, or some dimensions of the prediction space over others.

While there are many possible geometries and metric spaces that could be defined over prediction spaces, they are difficult to specify without more information. We focus on the metric space defined by the Mahalanobis norm $\|z\|_M := \left( z^\top M z \right)^{1/2}$, where $M$ is a positive semi-definite matrix. The Mahalanobis norm results in the prediction loss

$$\mathcal{L}_{\text{pred}}(\theta, \phi) := \mathbb{E}_{(x,y)\sim\mathcal{D}} \left[ \|f_\theta(x) - y\|^2_{\Lambda_\phi(x)} \right], \tag{3}$$

where $\Lambda_\phi$ is a metric parameterized by $\phi$ and this is also conditional on the feature $x$ so it can learn the importance of the regression space from each part of the feature space.

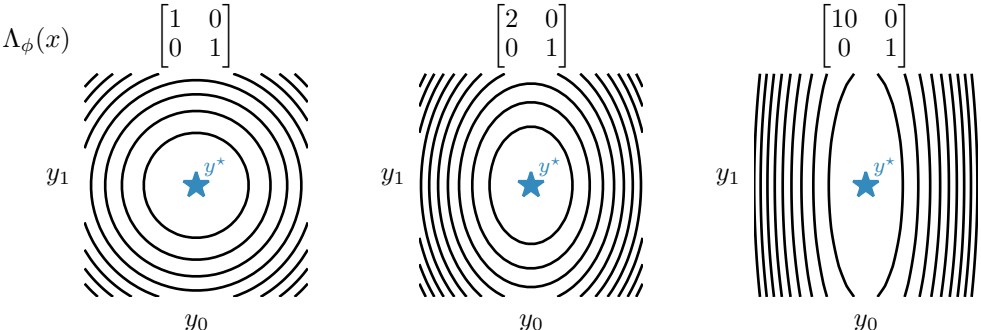

Figure 2: Examples of the Mahalanobis loss from eq. (3) in a 2-dimensional prediction task. The model's loss is zero only when $\hat{y} = y^\star$. Here, the metric $\Lambda_\phi(x)$ increases the weighting on the $y_0$ component of the loss and thus emphasizes the predictions along this dimension.

Many methods can be seen as hand-crafted ways of setting a Mahalanobis metric, including: 1) normalizing the input data by making the metric appropriately scale the dimensions of the prediction, 2) re-weighting the samples as in Donti et al. [2017], Lambert et al. [2020] by making the metric scale each sample based on some importance factor, or 3) using other performance measures, such as the value gradient in Voelcker et al. [2022].

More generally beyond these, the Mahalanobis metrics help emphasize the:

1. *relative importance of dimensions*. the metric allows for down- or up-weighting different dimensions of the prediction space by changing the diagonal entries of the metric. Figure 2 illustrates this.

2. *correlations in the prediction space*. the quadratic nature of the loss with the metric allows the model to be aware of correlations between dimensions in the prediction space.

3. *relative importance of samples*. heteroscedastic metrics $\Lambda(x)$ enable different samples to be weighted differently for the final expected cost over the dataset.

Without more information, parameterizing and specifying the best metric for learning the model is challenging as it involves the subproblem of understanding the relative importance between predictions. We suggest that when it is available, the downstream task information characterizing the overall model's performance can be used to learn a metric in the prediction space. Hence, learning model parameters with a metricized loss can be seen as conditioning the learning problem. The ability to learn the metric end-to-end enables the task to condition the learning of the model in any or all of the three ways described above. This approach offers an interpretable method for the task to guide the model learning, in contrast to relying solely on task gradients for learning model parameters, which may or may not align effectively with the given prediction task.

### 3.2 End-to-end metric learning for model learning

The key idea of the method is to learn a metric end-to-end with a given task, which is then used to train the prediction model as shown in eq. (3). The learning problem of the metric and model parameters are formulated as the bilevel optimization problem

$$\phi^\star := \arg\min_\phi \; \mathcal{L}_{\text{task}}(\theta^\star(\phi)), \tag{4}$$

$$\text{subject to} \;\; \theta^\star(\phi) = \arg\min_\theta \mathcal{L}_{\text{pred}}(\theta, \phi) \tag{5}$$

where $\phi$ and $\theta$ are (respectively) the metric and model parameters, $\mathcal{L}_{\text{pred}}$ is the metricized prediction loss (eq. (3)) to train the prediction model, and $\mathcal{L}_{\text{task}}$ is the task loss defined by the task at hand (which could be another optimization problem, e.g. eq. (8), or another learning task, e.g. eq. (10).

**Gradient-based learning.** We learn the optimal metric $\Lambda_{\phi^\star}$ with the gradient of the task loss, i.e. $\nabla_\phi \mathcal{L}_{\text{task}}(\theta^\star(\phi))$. Using the chain rule and assuming we have the optimal $\theta^\star(\phi)$ for some metric

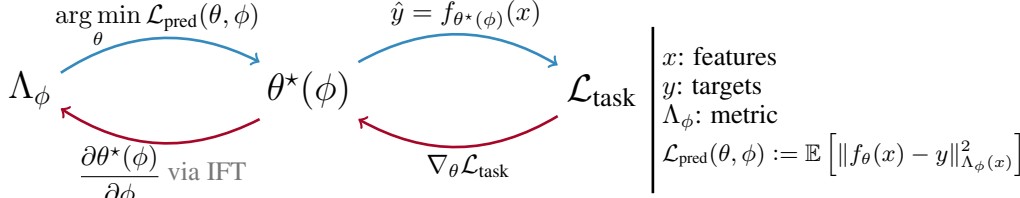

Figure 3: TaskMet learns a metric for predictions with the gradient from a downstream task loss.

---

**Algorithm 1** TaskMet: Task-Driven Metric Learning for Model Learning

---

**Models:** predictor $f_\theta$ and metric $\Lambda_\phi$ with initial parameterizations $\theta$ and $\phi$
**while** unconverged **do**
    *// approximate $\theta^\star(\phi)$ given the current metric $\Lambda_\phi$*
    **for** $i$ in $1 \ldots K$ **do**
        $\theta \leftarrow \text{update}(\theta, \nabla_\theta \mathcal{L}_{\text{pred}}(\theta, \phi))$ *// fit the predictor $f_\theta$ to the current metric loss* (eq. (3))
    **end for**
    $\phi \leftarrow \text{update}(\phi, \nabla_\phi \mathcal{L}_{\text{task}})$ *// update the metric $\Lambda_\phi$ with the task loss* (eq. (6))
**end while**
**return** optimal predictor $f_\theta$ and metric $\Lambda_\phi$ solving the bi-level problem in eq. (4)

---

parameterization $\phi$, this derivative is

$$\nabla_\phi \mathcal{L}_{\text{task}}(\theta^\star(\phi)) = \nabla_\theta \mathcal{L}_{\text{task}}(\theta)\big|_{\theta=\theta^\star(\phi)} \cdot \frac{\partial \theta^\star(\phi)}{\partial \phi} \tag{6}$$

To calculate the term $\nabla_\phi \mathcal{L}_{\text{task}}(\theta^\star(\phi))$, we need to compute two gradient terms: $\nabla_\theta \mathcal{L}_{\text{task}}(\theta)\big|_{\theta=\theta^\star(\phi)}$ and $\partial \theta^\star(\phi)/\partial \phi$. The former can be estimated in standard way since $\mathcal{L}_{\text{task}}(\theta)$ is an explicit function of $\theta$. However, the latter cannot be directly calculated because $\theta^\star$ is a function of optimization problem which is multiple iterations of gradient descent, as shown in eq. (5). Backpropping through multiple iterations of gradient descent can be computationally expensive, so we use the implicit function theorem (appendix A) on the first-order optimality condition of eq. (5), i.e. $\frac{\partial \mathcal{L}_{\text{pred}}(\theta, \phi)}{\partial \theta} = 0$. Combining these, $\nabla_\phi \mathcal{L}_{\text{task}}(\theta^\star(\phi))$ can be computed with

$$\nabla_\phi \mathcal{L}_{\text{task}}(\theta^\star(\phi)) = \nabla_\theta \mathcal{L}_{\text{task}}(\theta) \cdot \underbrace{- \left( \frac{\partial^2 \mathcal{L}_{\text{pred}}(\theta, \phi)}{\partial \theta^2} \right)^{-1} \frac{\partial^2 \mathcal{L}_{\text{pred}}(\theta, \phi)}{\partial \phi \partial \theta} \Bigg|_{\theta=\theta^\star(\phi)}}_{\partial \theta^\star / \partial \phi} \tag{7}$$

The implicit derivatives in eq. (7) may be challenging to compute or store in memory because the Hessian term $\partial^2 \mathcal{L}_{\text{pred}}(\theta, \phi)/\partial \theta^2$ is the Hessian of the prediction loss with respect to the model's parameters. Approaches such as Lorraine et al. [2020] are able to scale related implicit differentiation problems to models with millions of hyper-parameters. The main insight is that the Hessian does not need to be explicitly formed or inverted and the entire implicit derivative term needed for backpropagation can be obtained with an implicit solver. We follow Blondel et al. [2022] and compute the implicit derivative by using conjugate gradient on the normal equations.

## 4 Experiments

We evaluate our method in two distinct settings: 1) when the downstream task involves an optimization problem parameterized by the prediction model output, and 2) when the downstream task is another learning task. For the first setting, we establish our baselines by replicating experiments from previous works such as Shah et al. [2022] and Wilder et al. [2019]. These baselines encompass tasks like portfolio optimization and budget allocation. In the second setting, we focus on model-based

reinforcement learning. Specifically, we concentrate on learning a dynamics model (prediction model) and aim to optimize the Q-value network using the learned dynamics model for the Cartpole task [Nikishin et al., 2022]. Appendix B provides further experimental details and hyper-parameter information.

## 4.1 Metric parameterization

We parameterize the metric using a neural network with parameters $\phi$, denoted as $\Lambda_\phi := L_\phi^\top L_\phi$, where $L_\phi$ is an $n \times n$ matrix, where $n$ is the dimension of the prediction space. This particular factorization constraint ensures that the matrix is positive semi-definite, which is crucial for it to be considered a valid metric. The neural network parameters are initialized to make $\Lambda_\phi$ closer to the identity matrix $\mathbb{I}$, representing the Euclidean metric. The learned metric can be conditional on the input, denoted as $\Lambda_\phi(x)$, or unconditional, represented as $\Lambda_\phi$, depending on the problem's structure.

## 4.2 Decision-Focused Learning

### 4.2.1 Background and experimental setup

We use three standard resource allocation tasks for comparing task-based learning methods [Shah et al., 2022, Wilder et al., 2019, Donti et al., 2017, Futoma et al., 2020]. In this setting, resource utility prediction based on some input features constitute a prediction model, resource allocation constitutes the downstream task which is characterized by $\mathcal{L}_{\text{task}}$ The prediction model's output parameterized the downstream resource optimization. The settings are implemented exactly as in Shah et al. [2022] and have task losses defined by

$$\mathcal{L}_{\text{task}}(\theta) := \mathbb{E}_{(x,y)\sim\mathcal{D}}[g(z^\star(\hat{y}), y)] \tag{8}$$

where $z^\star(\hat{y}) := \arg\min_z g(z, \hat{y})$ and $g(z, y')$ is some combinatorial optimization objective over variable $z$ parameterized by $y'$. The task loss $\mathcal{L}_{\text{task}}$ is the expected value of objective function with decision variable $z^\star(\hat{y})$ induced by the prediction model $\hat{y} = f_\theta(x)$ under the ground truth parameters $y$. We use corresponding surrogate losses to replicate the $z^\star(\hat{y})$ optimization problem as in Shah et al. [2022], Wilder et al. [2019], Xie et al. [2020] and differentiate through the surrogate using `cvxpylayers` [Agrawal et al., 2019].

These settings evaluate the ability of TaskMet to capture the correlation between model predictions and differentiate between different data-points in accordance to their importance for the optimization problem. Hence, we consider a heteroscedastic metric, i.e., $\Lambda_\phi(x)$.

**Baselines.** We compare with standard baseline losses for learning models:

1. The standard MSE loss $\theta^\star = \arg\min_\theta \mathbb{E}_{(x,y)\sim\mathcal{D}}[(f_\theta(x) - y)^2]$. This method doesn't use any task information.
2. DFL [Wilder et al., 2019], which trains the prediction model with a weighted combination of $\mathcal{L}_{\text{task}}$ and $\mathcal{L}_{\text{pred}}$ as in eq. (1).
3. LODL Shah et al. [2022], which learns a parametric loss for each point in the training data to approximate the $\mathcal{L}_{\text{task}}$ around that point. That is, $LODL_{\psi_n}(\hat{y}_n) \approx \mathcal{L}_{\text{task}}(\hat{y}_n)$ for all $n$. They create a dataset of $\{(\hat{y}_n, \mathcal{L}_{\text{task}}(\hat{y}_n))\}$ for $\hat{y}_n$ sampled around the $y_n$. After this they learn the LODL loss for each point as $\psi_n^\star = \arg\min_{\psi_n} \frac{1}{K}\sum_{k=1}^K (LODL_{\psi_n}(y_n^k) - \mathcal{L}_{\text{task}}(y_n^k))^2$. We chose the "Quadratic" variant of their method which is the closest to ours, where $LODL_{\psi_n}(\hat{y}) = (\hat{y} - y)^\top \psi_n(\hat{y} - y)$ where $\psi_n$ is a learned symmetric Positive semidefinite (PSD) matrix. LODL also uses eq. (1) to learn the model parameters, but using $LODL_{\psi_n}(\hat{y}_n) \approx \mathcal{L}_{\text{task}}(\hat{y}_n)$

**Experimental settings.** We use the following experimental settings from [Shah et al., 2022]:

1. **Cubic**: This setting evaluates methods under model mismatch scenario where the model being learned suffers with severe approximation error. In this task, it is important for methods to allocate model capacity to the points more critical for the downstream tasks.
   *Prediction Model*: A linear prediction model $f_\theta(x) := \theta x$ is learned for the problem where the ground truth data is generated by cubic function, i.e., $y_i = 10x_i^3 - 6.5x_i, x_i \in U[-1, 1]$.
   *Downstream task*: Choose top $B = 1$ out of $M = 50$ resources $\mathbf{\hat{y}} = [\hat{y}_1, \ldots, \hat{y}_M]$, $z^\star(\mathbf{\hat{y}}) := \arg\max_i \mathbf{\hat{y}}$

Table 2: Normalized test decision quality (DQ) on the decision oriented learning problems.

| | | | Problems | |
|---|---|---|---|---|
| Method | $\alpha$ | Cubic | Budget | Portfolio |
| MSE | | $-0.96\pm0.02$ | $0.54\pm0.17$ | $0.33\pm0.03$ |
| DFL | 0 | $0.61\pm0.74$ | $0.91\pm0.06$ | $0.25\pm0.02$ |
| DFL | 10 | $0.62\pm0.74$ | $0.81\pm0.11$ | $0.34\pm0.03$ |
| LODL | 0 | $0.96\pm0.005$ | $0.84\pm0.105$ | $0.17\pm0.05$ |
| LODL | 10 | $-0.95\pm0.005$ | $0.58\pm0.14$ | $0.30\pm0.03$ |
| TaskMet | | $0.96\pm0.005$ | $0.83\pm0.12$ | $0.33\pm0.03$ |

0=random model 1=oracle model

Table 3: Test prediction errors (MSE) on the decision oriented learning problems.

| | | | Problems | |
|---|---|---|---|---|
| Method | $\alpha$ | Cubic | Budget $(\times 1e^{-4})$ | Portfolio $(\times 1e^{-4})$ |
| MSE | | $2.30\pm0.03$ | $4.32\pm2.35$ | $4.03\pm0.24$ |
| DFL | 0 | $2.89\pm0.32$ | $71.7\pm58.3$ | $8.0e^3\pm8e^2$ |
| DFL | 10 | $2.41\pm0.05$ | $8.09\pm12.1$ | $5.18\pm0.46$ |
| LODL | 0 | $2.88\pm0.03$ | $35.9\pm12.9$ | $55.6\pm9.95$ |
| LODL | 10 | $2.29\pm0.19$ | $5.05\pm1.88$ | $4.31\pm0.31$ |
| TaskMet | | $2.89\pm0.03$ | $9.74\pm13.79$ | $4.69\pm0.56$ |

$\alpha$ is the prediction loss weight in eq. (1)

2. **Budget Allocation**: Choose top $B = 2$ websites to advertise based on Click-through-rates (CTRs) predictions of $K$ users on $M$ websites.
   *Prediction Model*: $\hat{\mathbf{y}}_m = f_\theta(x_m)$ where $x_m$ is given features of $m^{\text{th}}$ website and $\hat{\mathbf{y}}_m = [\hat{y}_{m,1}, \ldots, \hat{y}_{m,K}]$ is the predicted CTRs for $m^{\text{th}}$ website for all $K$ users.
   *Downstream task*: Determine $B = 2$ websites such that the expected number of users that click on the ad at least once is maximized, i.e., $z^\star(\hat{\mathbf{y}}_m) = \arg\max_z \sum_{j=0}^{K}(1-\prod_{i=0}^{M} z_i \cdot \hat{y}_{ij})$ where $z_i \in \{0,1\}$.

3. **Portfolio Optimization**: The task is to choose a distribution over $M$ stocks in Markowitz portfolio optimization [Markowitz and Todd, 2000, Michaud, 1989] that maximizes the expected return under the risk penalty.
   *Prediction Model*: Given the historical data $x_m$ about a stock $m$, predict the future stock price $\hat{y}_m$. Combining prediction over $M$ stocks to get $\hat{\mathbf{y}} = [\hat{y}_1, \ldots, \hat{y}_M]$.
   *Downstream Task*: Given the correlation matrix $Q$ of the stocks, choose a distribution over stocks $\mathbf{z}^\star(\hat{\mathbf{y}}) = \arg\max_{\mathbf{z}} \mathbf{z}^\top\hat{\mathbf{y}} - \lambda\mathbf{z}^\top Q\mathbf{z}$ s.t. $\sum_{i=0}^{M} z_i \leq 1$   and   $0 \leq z_i \leq 1, \forall i$

We run our own experiments for LODL [Shah et al., 2022] using their public code.

#### 4.2.2 Experimental results

Table 2 presents a summary of the performance of different methods on all the tasks. Each problem poses unique challenges for the methods. The *cubic* setting suffers from severe approximation errors, hence the learning method needs to allocate limited model capacity more towards higher utility points compared to lower utility points. The MSE method performs the worst as it lacks task information and only care about prediction error. DFL with $\alpha = 0$ performs better than MSE, but it can get trapped in local optima, leading to higher variance in the problem [Shah et al., 2022]. LODL ($\alpha = 0$) performs among the highest in this problem since it uses learned loss for each point. TaskMet performs as good as LODL as it can capture the relative importance of higher utility points versus lower utility points using the learned metric, resulting in more accurate predictions for those points (see fig. 4). In *budget allocation*, DFL (with $\alpha = 0$) performs the best, since it is solely optimizer over $\mathcal{L}_{\text{task}}$, but on the other hand it has 10 orders of larger prediction error as shown in table 3 indicating that the model is overfit to the task, LODL ($\alpha = 0$) suffers from the same problem. TaskMet has the $2^{\text{nd}}$ best Decision

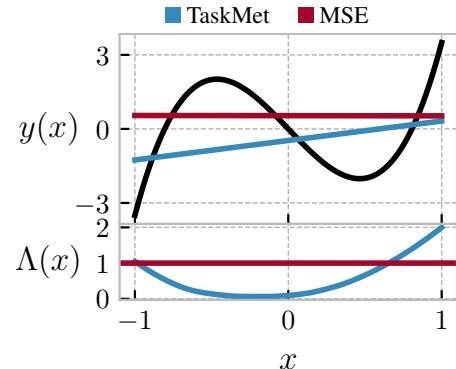

Figure 4: (Cubic problem) TaskMet learns a metric that prioritizes points that are the most important the downstream task. The euclidean metric (MSE) puts equal weight on all points and leads to a bad model with respect to the downstream task.

Quality without overfitting on the task, i.e., low prediction error. In *Portfolio Optimization*, the decision quality correlates highly with the model accuracy/prediction error as in this setting the optimization problem mostly depends upon the accurate prediction of the stocks. This is the reason that MSE, DFL ($\alpha = 10$) performs the best, but DFL ($\alpha = 0$) performs the worst, since it has solely

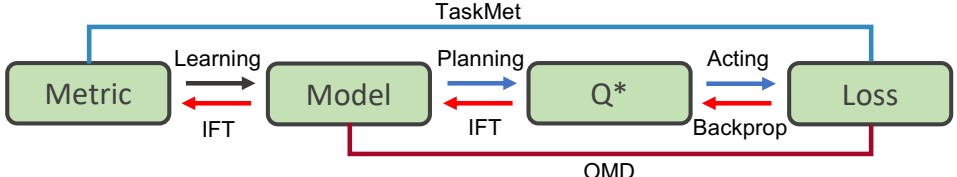

Figure 5: OMD [Nikishin et al., 2022] uses the planning task loss to learn the model parameters using implicit gradients. TaskMet add one more optimization step over OMD and instead of learning the model parameters using task loss, we learn the metric which then is used to learn model parameters.

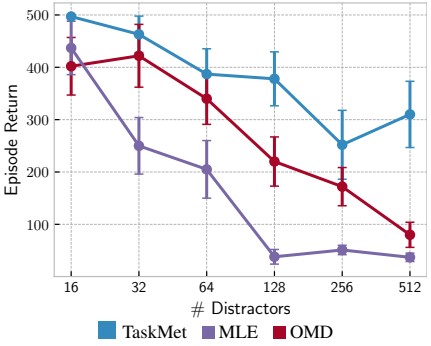

Figure 6: Results on the cartpole with distracting states [Nikishin et al., 2022].

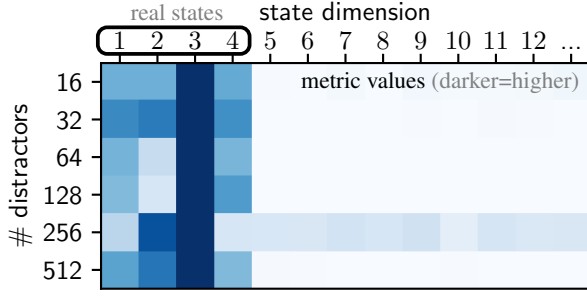

Figure 7: Our learned metric successfully distinguishes the real states from the distracting states, i.e. the real states take a higher metric value.

being trained on $\mathcal{L}_{\text{task}}$ without any $\mathcal{L}_{\text{pred}}$. As shown in table 2 and table 3, TaskMet is the only method that consistently performs well considering both task loss and prediction loss, across all the problem settings, this is due to the ability of the metric to infer problem-specific features without manual tuning, unlike other methods.

## 4.3 Model Based Reinforcement Learning

### 4.3.1 Background and experimental setup

Model-based RL suffers from objective-mismatch [Bansal et al., 2017, Lambert et al., 2020]. This is because dynamics models trained for data likelihood maximization do not translate to optimal policy. To reduce objective-mismatch, different losses [Farahmand et al., 2017, Voelcker et al., 2022] have been proposed to learn the model which is better suited to learning optimal policies. TaskMet provides an alternative approach towards reducing objective-mismatch, as the prediction loss is directly learnt using task loss. We set up the MBRL problem as follows. Given the current state $s_t$ and control $a_t$ at a timestep $t$ of a discrete-time MDP, the *dynamics model* predicts the next state transition, i.e. $\hat{s}_{t+1} := f_\theta(s_t, a_t)$. The prediction loss is commonly the squared error loss $\mathbb{E}_{s_t, a_t, s_{t+1}} \|s_{t+1} - f_\theta(s_t, a_t)\|_2^2$, and the downstream task is to find the optimal Q-value/policy. Nikishin et al. [2022] introduces idea of *optimal model design* (OMD) to learn the dynamics model end-to-end with the policy objective via implicit differentiation. Let $Q_\omega(s, a)$ be the action-conditional value function parameterized by $\omega$. The Q network is trained to minimize the Bellman error induced by the model $f_\theta$:

$$\mathcal{L}_Q(\omega, \theta) := \mathbb{E}_{s,a}[Q_w(s, a) - \mathrm{B}^\theta Q_{\bar{w}}(s, a)]^2, \tag{9}$$

where $\bar{\omega}$ is moving average of $\omega$ and $\mathrm{B}^\theta$ is the model-induced Bellman operator $\mathrm{B}^\theta Q_{\bar{w}}(s, a) := r_\theta(s, a) + \gamma \mathbb{E}_{p_\theta(s,a,s')}[\log \sum_{a'} \exp Q(s', a')]$. Q-network optimality defines $\omega$ as an implicit function of the model parameters $\theta$ as $\omega^\star(\theta) = \arg\min_\omega \mathcal{L}_Q(\omega, \theta) \implies \frac{\partial \mathcal{L}_Q(\omega, \theta)}{\partial \omega} = 0$. Now we have task loss which is optimized to find optimal Q-values:

$$\mathcal{L}_{\text{task}}(\omega^\star(\theta)) := \mathbb{E}_{s,a}[Q_{\omega^\star(\theta)}(s, a) - \mathrm{B}Q_{\bar{\omega}}(s, a)]^2 \tag{10}$$

where the Bellman operator induced by ground-truth trajectory and reward is $\mathrm{B}Q_{\bar{\omega}}(s, a) := r(s, a) + \gamma \mathbb{E}_{s,a,s'} \log \sum_{a'} \exp Q_{\bar{\omega}}(s', a')$.

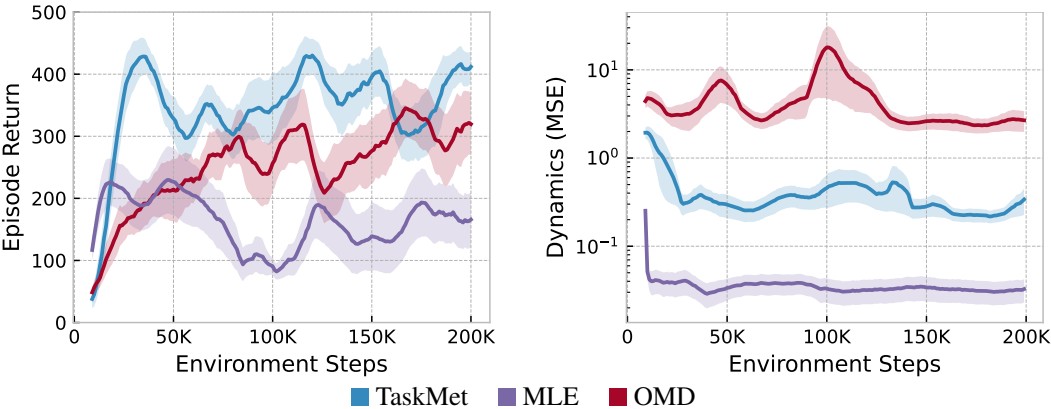

Figure 8: Results on cartpole with a reduced model capacity from Nikishin et al. [2022].

**OMD setup.** OMD end-to-end optimizes the model for the task loss, i.e. $\theta^\star = \arg\min_\theta \mathcal{L}_{\text{task}}(\omega^\star(\theta))$.

**TaskMet setup.** For metric learning, we extend OMD to learn a metric using task gradients, to train the model parameters, see fig. 5. Metric learning just adds one more level of optimization to OMD and results in the *tri-level* problem

$$\phi^\star = \arg\min_\phi \ \mathcal{L}_{\text{task}}(\omega^\star)$$

$$\text{subject to} \ \ \omega^\star(\theta^\star) = \arg\min_\omega \mathcal{L}_Q(\omega, \theta^\star) \tag{11}$$

$$\theta^\star(\phi) = \arg\min_\theta \mathcal{L}_{\text{pred}}(\phi, \theta)$$

where $\mathcal{L}_{\text{task}}(\omega^\star)$ and $\mathcal{L}_Q(\omega, \theta^\star)$ are defined in eq. (10) and eq. (9), respectively, and $\mathcal{L}_{\text{pred}}(\phi, \theta) = \mathbb{E}_{s_t, a_t, s_{t+1}} \|s_{t+1} - f_\theta(s_t, a_t)\|^2_{\Lambda_\phi(s_t)}$ is the metricized prediction loss in eq. (3).

To learn $\phi^\star$ using gradient descent, we estimate $\nabla_\phi \mathcal{L}_{\text{task}}$ as

$$\nabla_\phi \mathcal{L}_{\text{task}} = \nabla_\omega \mathcal{L}_{\text{task}}(\omega^\star) \cdot \frac{\partial \omega^\star}{\partial \theta^\star} \cdot \frac{\partial \theta^\star}{\partial \phi}$$

$$= \nabla_\omega \mathcal{L}_{\text{task}}(\omega^\star) \cdot \left(\frac{\partial^2 \mathcal{L}(\omega, \theta^\star)}{\partial \omega^2}\right)^{-1} \cdot \left.\frac{\partial^2 \mathcal{L}(\omega, \theta^\star)}{\partial \theta \partial \omega}\right|_{\omega^\star(\theta^\star)} \cdot \left(\frac{\partial^2 \mathcal{L}_{\text{pred}}(\theta, \phi)}{\partial \theta^2}\right)^{-1} \cdot \left.\frac{\partial^2 \mathcal{L}_{\text{pred}}(\theta, \phi)}{\partial \phi \partial \theta}\right|_{\theta^\star(\phi)} \tag{12}$$

#### 4.3.2 Experimental results

We replicated experiments from Nikishin et al. [2022] on the Cartpole environment. The first experiment involved state distractions, where the state of the agent was augmented with noisy and uninformative values. In this setting, we considered an unconditional diagonal metric of dimension $n$, which is the dimension of the state space, i.e. $\Lambda_\phi := \text{diag}(\phi)$, where $\phi \in \mathbb{R}^n$. As shown in fig. 6, the MLE method performed the worst across different numbers of distracting states, as it allocated its capacity to learn distracting states as well. TaskMet outperformed the other methods in all scenarios. The superior performance of TaskMet with distracting states can be attributed to the metric's ability to explicitly distinguish informative states from noise states using the task loss and then train the model using the given metric, as shown in fig. 7. The learned metric in fig. 7 assigned the highest weight to the third dimension of the state space, which corresponds to the pole angle — the most indicative dimension for the reward. This shows that the metric can differentiate state dimensions based on their importance to the task.

We also consider a setting with reduced model capacity, where the network is under-parametrized, forcing the model to prioritize how it allocates its capacity. In this scenario, we employ a full conditional metric, denoted as $\Lambda_\phi = \Lambda_\phi(x)$, which enables the metric to weigh dimensions and state-action pairs differently. We conducted the experiment using a model size of 3 hidden units in the layer. As depicted in fig. 8, TaskMet achieves a better return on evaluation compared to MLE and OMD. Additionally, it is evident that TaskMet achieves a lower MSE on the model predictions compared to OMD, indicating that learning with the metric contributes to a better dynamics model.

# 5 Conclusion and discussion

In conclusion, this paper addresses the challenge of combining task and prediction losses in task-based model learning. While task-based learning methods offer the advantage of discovering task-relevant features and data samples without explicit inductive biases, the current trend of using task loss alongside prediction loss has potential limitations. These limitations include overfitting of the prediction model to a specific task, rendering it ineffective for other tasks, and the lack of interpretability in the task-relevant features learned by the prediction model.

To overcome these limitations, the paper introduces the concept of task-driven metric learning, which integrates the task loss into a parameterized prediction loss. This approach enables end-to-end learning of metrics to train prediction models, allowing the models to focus on task-relevant features and dimensions in the prediction space. Moreover, the resulting prediction models become more interpretable, as metric learning serves as a preconditioning step for gradient-based model training. The effectiveness of the method is shown using different scales of experimental setting - decision oriented tasks as well as downstream learning tasks.

One of the limitations of the method is stability of learning the metric. Bad gradients can lead collapsed metric which can lead to unrecoverable bad predictions. Hence, hyper-parameter tuning of learning rate for metric learning and parameterization choices of the metric are crucial. Possible extensions to this work includes end-to-end metric learning with multiple task losses, learning metric for training dynamics models to be used for long-horizon planning tasks, etc.

### Acknowledgments

We would like to thank Brian Karrer, Claas Voelcker, Karen Ullrich, Leon Bottou, Maximilian Nickel, and Mike Rabbat insightful comments and discussions.

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

# A The implicit function theorem

We used the implicit function theorem to compute the derivative of the prediction model with respect to the metric's parameters in eq. (7). For completeness, this section briefly presents the standard implicit function theorem, *cf.* Dini [1878] and Dontchev and Rockafellar [2009, Theorem 1B.1]:

**Theorem 1 (Implicit Function Theorem)** *Implicit Function Theorem: Let $f : \mathbb{R}^n \times \mathbb{R}^m \to \mathbb{R}$ be a continuous differentiable function, and let $x^\star, y^\star$ be a point satisfying $f(x^\star, y^\star) = 0$. If the Jacobian $\frac{\partial(f(x^\star, y^\star))}{\partial y} \neq 0$, then there exists an open set around $(x^\star, y^\star)$ and a unique continuously differentiable function $g$ such that $y^\star = g(x^\star)$ and $f(x, g(x)) = 0$. Additionally, the following relation holds:*

$$\frac{\partial g(x)}{\partial x} = -\left( \frac{\partial f(x, y^\star)}{\partial y} \right) \frac{\partial f(x, y^\star)}{\partial x}\big|_{y^\star = g(x)} \tag{13}$$

# B Implementation Details

## B.1 Decision Oriented Model Learning

We replicated our experiments using the codebase provided by Shah et al. [2022], which can be found on github. To ensure consistency, we used the same hyperparameters as mentioned in the code or article for the baselines. Our metric learning pipeline was added on top of their code, and thus we focused on tuning hyperparameters related to metric learning. The metric is parameterized as $\Lambda_\phi(x) = L_\phi(x)L_\phi^\top(x) + \epsilon_\phi \mathbb{I}_{n \times n}$, where $\epsilon_\phi$ is a learnable parameter that explicitly controls the amount of Euclidean metric in the predicted metric. This helps ensure the stability of metric learning. We initialize the parameters in such a way that the predicted metric is close to the Euclidean metric. For each outer loop of metric update, we perform $K$ inner updates to train the predictor. Following the methodology of Shah et al. [2022], we conducted 50 runs with different seeds for each of the experiments, where each method was evaluated on 10 different datasets, with 5 different seeds used for each dataset.

Table 4: Hyper-parameters for Decision Oriented Learning Experiments

| Hyper-Parameter | Values |
|---|---|
| $\Lambda_\phi$ learning rate | $10^{-3}$ |
| $\Lambda_\phi$ hidden layer sizes | [200] |
| Warmup steps | 500 |
| Inner Iterations ($K$) | 100 |
| Implicit derivative batchsize | 10 |
| Implicit derivative solver | Conjugate gradient on the normal equations (5 iterations) |

## B.2 Model-Based Reinforcement Learning

We consider the work of Nikishin et al. [2022] as the baseline for replicating the experiments, and we build upon their source code. Our metric learning is just one additional step to their method. We adopt exact same hyperparameters as their for dynamics learning and Action-Value function learning. We focus on exploring and tuning the hyper-parameters specific to the metric learning component of the method.

Table 5: Hyper-parameters for the CartPole experiments

| Hyper-Parameter | Values |
|---|---|
| $\Lambda_\phi$ learning rate | $10^{-3}$ |
| $\Lambda_\phi$ hidden layer sizes | [32, 32] |
| Warmup steps | 5000 |
| Inner iterations ($K$) | 1 |
| Implicit derivative batchsize | 256 |
| Implicit derivative solver | Conjugate gradient on the normal equations (10 iterations) |

For the state distractor experiments, we parameterize the metric as an unconditional diagonal matrix, denoted as $\Lambda_\phi = \text{diag}(\phi)$ where $\phi \in \mathbb{R}^n$ and $n$ is the dimension of the state space. In addition, we consider a hyper-parameter of *metric parameterization*, for which we either take normalize or unnormalized metric. When we refer to normalizing the metric, we mean constraining the norm of the $\phi$ vector to be equal to the L2 norm of an euclidean metric which is used by MSE method. This constrains the family of learnable metrics. To achieve this, we set $\phi := \frac{\phi}{\|\phi\|_2}\sqrt{n}$, ensuring $\|\phi\|_2 = \|\mathbb{I}_{n \times n}\|_2 = \sqrt{n}$. We also used L1 regularization on the metric output, to induce sparsity in the metric. We sweep over three values of the regularization coefficient - $[0.0, 0.001, 0.1]$. We ran a sweep over the 6 combinations of hyperparameters - $[\text{unnormalized}, \text{normalized}] \times [0.0, 0.001, 0.1]$ and choose the best hyper-parameter combination for each of the experiment. All the number reported in the experiments are calculated over 10 random seeds.

Our metric learning approach uses two implicit gradient steps. Firstly, we take the implicit derivative through action-value network parameters, approximating the inverse hessian to the identity, similar to Nikishin et al. [2022]. Secondly, for the step through dynamics network parameters, we calculate the exact implicit derivative.

