# OpenReview forum: "TaskMet: Task-driven Metric Learning for Model Learning"
_NeurIPS.cc/2023/Conference — NeurIPS 2023 poster_

### Official Review · Reviewer_wsqn · 2023-07-06

**Soundness:** 3 good
**Presentation:** 3 good
**Contribution:** 3 good
**Rating:** 7
**Confidence:** 4

**Summary:**

The authors propose using task information to modify a metric that the prediction model uses for training and include task information. This way, they encourage the model to generate more accurate predictions specifically for inputs relevant to the downstream task.

**Strengths:**

 - The manuscript is well-structured, and the subject of research is relevant
 - The authors provide an introduction referencing a comprehensive set of related work

**Weaknesses:**

We have not identified strong weaknesses in this paper.

**Questions:**

GENERAL COMMENTS:
 - (1) Normalized Test Decision Quality: please provide an equation and reference this metric.
 - (2) The authors claim that one of the key contributions is a "more interpretable learning of the model using the metric compared to learning with a combination of task loss and prediction loss." Nevertheless, there is no comparison or assessment on how/how much more is the proposed approach more interpretable.

TABLES:
 - (3) Table 1: align reported results to the right. Ensure they all have the same number of decimals.
 - (4) Table 2: same comment as for Table 1. Add arrows (up/down) beside the metric name to indicate whether higher/lower is better.

FIGURES:
 - (5) Figure 3: provide more spacing between both Figures to clarify that they refer to different analyses. Furthermore, both can be referenced as (a) and (b). Moreover, there is no reference for the black curve at the top. Provide some short explanation of both Figures in the caption.
 - (6) Figure 6: add a brief interpretation of the results.

SPELLING/WORDING:
 - (7) "parametrize generalized mahalanobis distance based loss for training the prediction models" -> "parametrize generalized Mahalanobis distance based loss for training the prediction models"
 - (8) "value network learning insteead of minimizing" -> value network learning instead of minimizing
 - (9) "Given the correlatoin matrix Q" ->  "Given the correlation matrix Q"
 - (10) "We use three resource allocation tasks that has been used" -> "We use three resource allocation tasks that have been used"
 - (11) "Budget Allcoation can be found" -> "Budget Allocation can be found"
 - (12) "value network learning insteead of minimizing" -> "value network learning instead of minimizing"
 - (13) "fig. X" -> "Fig. X"
 - (14) "eq. X" -> "Eq. X"
 - (15) "table X" -> "Table X"
 - (16) "the ability of metric to infer problem specific features without manual tuning unlike other methods" -> "the ability of the metric to infer problem-specific features without manual tuning unlike other methods"
 - (17) "underlying properties of the task for useful for training the model." -> "underlying properties of the task useful for training the model."

**Limitations:**

The authors have identified and acknowledged limitations of the proposed methods.

---

> ### Author Rebuttal · Authors · 2023-08-09
>
> Thank you for the review of our paper and detailed comments. We are happy to hear you enjoyed the paper. Thanks also for  pointing out the other minor problems about the typos and figures. We will correct them. Here are some clarifications to your comments, please let us know if there is anything else.
>
> > (1) Normalized Test Decision Quality: please provide an equation and reference this metric.
>
> This is the same metric which is used by the baseline method in previous work [Shah et al., 2022]. Test Decision Quality is the negative of test task loss. Normalized Test Decision Quality basically means normalizing the decision quality such that decision quality of random prediction is mapped to 0 whereas decision quality of oracle predictions is mapped to 1. Hence the normalized DQ is given by $DQ_{normalized}=(DQ-DQ_{random})/(DQ_{oracle}-DQ_{random}) \in (-\infty, 1]$
>
> > (2) The authors claim that one of the key contributions is a "more interpretable learning of the model using the metric compared to learning with a combination of task loss and prediction loss." Nevertheless, there is no comparison or assessment on how/how much more is the proposed approach more interpretable.
>
> When we mentioned the interpretability of TaskMet, we were thinking about (1) the prediction model making predictions in the original space rather than a latent space as in DFL, and (2) the meaning of the metric’s values across parts of the prediction space. We realize now that our use of the word “interpretability” could have other meanings, such as somehow having better interpretations of *why* the model is making certain predictions, but we will clarify that this is not what we mean.
>
> For #2, the learned metric reveals inherent properties of the task. For example, in Figure 5, the metric puts the largest weight on dimension 3 in all the experiments, indicating that dimension 3 is the most important feature for the task. This is aligning very well with the fact that dimension 3 is actually pole angle, which is the most indicative feature for the cumulative reward since the task is to keep the pole straight. Similarly, in Figure 3, the learned metric, reveals the important prediction space for the task, allowing for the higher metric value at extreme high utility  ends, indicating that these samples should be priorities for accuracy as they are more critical for the downstream task, which makes sense as we look at the optimization problem we will see that the higher utility samples are the one which downstream task is more sensitive to.

---

> > ### Comment · Reviewer_wsqn · 2023-08-17
> >
> > We thank the authors for the detailed response. We consider they have addressed all of our comments.

---

### Official Review · Reviewer_jBQF · 2023-07-06

**Soundness:** 3 good
**Presentation:** 2 fair
**Contribution:** 3 good
**Rating:** 5
**Confidence:** 3

**Summary:**

The authors propose an approach to loss re-shaping for a supervised prediction model with the goal of improving the performance of a larger system that performs a downstream task using the predictions of the prediction model. Such shaping is necessary because reducing loss on the prediction task may only roughly correspond to improved performance on the downstream task. The authors show that their approach mostly outperforms standard maximum likelihood estimation and achieves similar performance to existing baselines.

**Strengths:**

TaskMet is an interesting method that enables injection of knowledge about downstream tasks without ignoring the potentially useful training data for the base prediction task.

**Weaknesses:**

The clarity of the presentation could be significantly improved. For example, the distinction between "task learning" and "prediction" is not clear to readers who are not very familiar with the terminology; these are interchangeable synonyms in many fields. The descriptions of the benchmark tasks are also not very accessible to those not already familiar with them (what are "resources" in the Cubic problem?).

The experimental results in section 5.2 are relatively weak; TaskMet does not show improved performance in any of the three benchmark tasks. Although the average performance across tasks is slightly better than existing approaches, it's not clear if TaskMet outperforms the baselines if they are well-tuned (and the authors note that hyperparameter tuning for TaskMet is key).

In section 5.3 (the model-based RL experiments), the results are stronger. However, the setting is somewhat contrived, in that the authors study either a) a large number of artificial noise dimensions added to the state, which is likely to improve TaskMet's performance because it is specifically designed to rescale the loss of each dimension and b) when the network is unrealistically small (3 hidden units). Although the results in these settings are positive, they do not decisively show the benefit of the proposed method in real-world settings. Further, the baselines considered are either only using the prediction task loss (MSE) or only using the downstream loss (OMD). It's not clear if TaskMet outperforms other methods that utilize both objectives.

Some related work could be added, particularly work in meta-learning loss functions, but this is not critical.

The advantages of TaskMet re: interpretability are not actually validated experimentally, other than on a synthetic problem showing that TaskMet learns to ignore dimensions of the targets that are artificial pure noise.

**Questions:**

Does TaskMet scale to more real-world settings, for example more difficult RL problems?

Does the learning metric produce useful information for real prediction problems, other than the artificial cartpole with noise setting?

Is TaskMet limited to learning a Mahalanobis norm? Why not a more complex/expressive learned loss function?

Does TaskMet require significantly more compute than MSE?

**Limitations:**

Yes, although addressing the differences in computational requirements (if any) would be helpful.

---

> ### Author Rebuttal · Authors · 2023-08-09
>
> Thank you for the detailed review and comments on our paper. We were surprised by your decision to reject the paper based on the clarity and experimental weaknesses, as we have evaluated the idea of learning a metric on standard experimental settings in the most recent DFL and end-to-end model learning papers. We do agree that these experimental settings are relatively small-scale, but they are standard in the DFL/task-based learning communities and we chose them so that we could focus our contribution only on adding metric learning to existing experimental settings. Even though small-scale, these tasks/experiments are representative of challenges present in task-based model learning paradigm. In light of this information, we hope you will agree that the work is at a stage ready to disseminate to the community and that you will consider raising your score to help with this. We address your specific questions below.
>
> > The clarity of the presentation could be significantly improved [...]
>
> We will distinguish between prediction and task loss for the camera-ready version. We have cited the papers from which we introduced these benchmarks, we have tried our best to mention all details about the benchmark tasks within the confines of the page constraints. Additionally, further details can be checked in prior works, e.g. we used the exact same benchmark as used in Shah et al [2022] and Nikishin et al. [2022]. Cubic problem is a toy experiment, we refer model outputs as resources in the example, as it’s consistent with the portfolio optimization and budget allocation terminology and enables us to ground the cubic example in a practical sense.
>
>
> > The experimental results in section 5.2 are relatively weak; [...]
>
> The goal in these demonstrations was to show that metric learning works well for standard DFL tasks while retaining better prediction accuracy on the tasks. While Table 1 shows that TaskMet’s end-to-end loss is similar to the DFL’s loss, Table 2 shows that TaskMet retains far better prediction accuracy than the model learned with the DFL, where the predictive model loses the ability to make predictions in the original space. We hope this way of thinking about using the end-to-end signal for the loss instead of the model will be a helpful shift for the larger settings DFL is used where it is important to keep the predictions in the original space, and we think that the idea of a metricized loss is far more reliable than just optimizing the DFL+MSE loss as in eq (1).
>
> Furthermore, the baselines are well-tuned in this experiment setting. In fact we reran the hyperparam sweep over the baselines to get best performance as the default parameters in their official codebase were not giving optimal performance for the baselines. Also, in this experiments, hyperparameter tuning is not very critical for TaskMet. That was a remark for model based RL setting.
>
> > In section 5.3 (the model-based RL experiments), the results are stronger. However, [...]
>
> We replicated the benchmark which has been used in previous works. We did artificial noise dimension according to the experiments done in the previous work [Nikishin et al. [2022] for fair comparison. Even though artificially injected noise, the experiments show that metric learns to distinguish between task relevant and irrelevant features. We agree that an MLP with 3 hidden units is small, but Nikishin et al. [2022] uses a hidden size 3; We chose this setting so we could directly compare our experimental results to OMD. We understand desire to see TaskMet's benefits in more complex settings. We share this enthusiasm and are actively exploring scaling TaskMet to challenging MDPs, and we will ensure that future work addresses this.
>
> > Some related work could be added, particularly work in meta-learning loss functions [...]
>
> We agree and will add references to similar meta learning loss functions, such as [this paper](https://arxiv.org/abs/1906.05374) on meta-learning a neural network as the loss. Please let us know if there are any others you think are relevant.
>
> > The advantages of TaskMet re: interpretability are not actually validated [...]
>
> Due to character constraints, we ask to please refer to **$2^{nd}$ point of our reply to Reviewer wsqn.**
>
> > Does TaskMet scale to more real-world settings, for example more difficult RL [...]
>
> Even though our experimental settings are relatively small-scale, they are standard settings in the field, e.g. as in Shah et al. [2022], that gave us a testbed for quantifying the impact of learning a metric. The datasets and tasks are reflective for what scaled versions of the tasks would look like: for budget allocation the dataset was real-world Yahoo! Website dataset, as well as the portfolio optimization is also on real stock data from 2004 to 2017 available with QuandlWIKI dataset. Training predictive models for stock prices from QuandlWIKI or budget allocations from the Yahoo! Website are already challenging tasks and there are a lot of tradeoffs that a model can make related to the types of errors to emphasize or parts of the prediction space to be more accurate. Our perspective is that the learned metric is capable of encoding some of this information.
>
> Furthermore, the cartpole setting is taken from OMD and a first step towards scaling task-based dynamics learning to larger MDPs. Our learned metric was not only able to distinguish between artificial noise but also produce useful information such as finding the important features for the task which in this case was pole angle.
>
> > Is TaskMet limited to learning a Mahalanobis norm? Why not a more complex/expressive learned loss function?
>
> Please refer to **$2^{nd}$ point of our reply to reviewer uXLc.**
>
> > Does TaskMet require significantly more compute than MSE?
>
> TaskMet requires minor extra compute compared to MSE. We have benchmarked this in terms of compute time per gradient step of TaskMet vs MSE. Details can be found in supplementary section B.

---

> > ### Comment · Reviewer_jBQF · 2023-08-19
> >
> > I appreciate the authors' clarifications about the experimental settings, usage of Mahalanobis norm, and tuning of the baselines. These have addressed some of my primary concerns.
> >
> > My primary remaining concern is the experimental settings. Although they may have been used in prior work, at some point, experimental settings must adapt to the complexity and capabilities of our methods, and the settings considered in this paper feel very simple and synthetic to me. Nonetheless, while I think the paper has shortcomings in evaluations, I would not be upset if it were accepted. Therefore I will raise my score to 5.

---

> ### Author Response · Authors · 2023-08-18
>
> Dear reviewer,
>
> Thank you again for taking the time to go through our paper. We've taken your comments into account in the latest (and hopefully clearer) version of our paper. When you get a chance, can you please read through our response and let us know if you are curious in discussing anything else?
>
> We still agree that the experimental settings generally studied by the DFL/OMD literature are relatively small-scale relative to other parts of the ML field, but scoped to this sub-field, our experimental settings are standard. We think our results sufficiently demonstrate and validate the idea of learning a metric rather than the model using the end-to-end loss. Can you please take this into consideration for your final evaluation of our paper?
>
> Best regards,
>
> The authors.

---

### Official Review · Reviewer_WKyg · 2023-07-07

**Soundness:** 3 good
**Presentation:** 2 fair
**Contribution:** 3 good
**Rating:** 6
**Confidence:** 3

**Summary:**

The paper proposes a novel method, called TaskMet, for learning a metric that can be used to improve the performance of prediction models on downstream tasks. The key idea is to use the task loss to guide the learning of the metric. This is done by using a gradient-based approach to optimize the metric parameters. The authors show that TaskMet can improve the performance of prediction models on a variety of tasks, including portfolio optimization, budget allocation, and model-based reinforcement learning.

**Strengths:**

+ The paper is well-written and easy to follow.
+ Motivations are clear and reasonable.
+ The proposed method is novel and has the potential to be widely applicable.
+ The authors provide extensive experimental results to support their claims.

**Weaknesses:**

It is not clear how sensitive the results are to the choice of hyperparameters. Please provide a more detailed analysis of the hyperparameter sensitivity.

**Questions:**

- Authors mentioned that hyper parameter tuning is important for metric learning. I was wondering what is the significance of the hyperparameters that were used in the experiments? Is this the reason for quite high standard deviations shown in Table 2 for TaskMet on Budget Allocation?
- Are the baselines results re-implemented by the authors? how did you pick hyper parameters for them?


**Limitations:**

Authors adequately addressed the limitations.

---

> ### Author Rebuttal · Authors · 2023-08-09
>
> Thanks for the review of our paper. We’re happy you found the paper well-written, well-motivated, and easy to follow. Here are some clarifications about the hyperparameters and baselines. To further help with reproducibility in the field we will also publish the entire source code necessary to reproduce every part of our paper.
>
> > It is not clear how sensitive the results are to the choice of hyperparameters. Please provide a more detailed analysis of the hyperparameter sensitivity.
>
> The mentioned sensitivity of hyperparameters was due to their usage in reinforcement learning experiments. As we know that reinforcement learning methods need careful hyperparameter tuning to work, hence it was the combination of TaskMet with RL which caused hyperparameter sensitivity. When using TaskMet for Decision focused model learning experiments, we didn’t find any particular sensitivity to hyperparameters. We didn’t perform any meaningful hyperparameter sensitivity analysis as it was not something core to the introduced method but rather something application/task oriented.
>
> > Authors mentioned that hyper parameter tuning is important for metric learning. I was wondering what is the significance of the hyperparameters that were used in the experiments? Is this the reason for quite high standard deviations shown in Table 2 for TaskMet on Budget Allocation?
>
> The significance of hyperparameters tuning in TaskMet is similar to what of other bilevel optimization problems. For example, number of inner iterations when solving for model parameters is critical, as to employ implicit gradient theorem we have to make sure that the inner optimization has converged, i.e,$ \nabla_\theta L(\theta, \phi)|_{\theta = \theta^\star(\phi)} = 0$. Other details about hyper-parameter can be found in supplementary material section A. The high standard deviation in Table 2 on Budget Allocation task is not due to hyper-parameters tuning of TaskMet. All other methods also have similar standard deviation, the higher standard deviation on this task is due the particular nature of task and dataset being used to train and test.
>
> > Are the baselines results re-implemented by the authors? how did you pick hyper parameters for them?
>
> Baselines of decision focused model learning were not re-implemented but were reproduced using the code and hyper-parameters provided by the original paper. We were in consultation with the author of the baseline to ensure fair reproduction of results. For model based RL experiments, we directly used the numbers reported in the original paper.

---

> > ### Comment · Reviewer_WKyg · 2023-08-17
> >
> > I'd like to thanks the authors for providing clarifications on the raised points. Having reviewed both the authors' response and the feedback provided by other reviewers, I find that all of my concerns have been adequately addressed. As a result, I am maintaining my original recommended score.

---

### Official Review · Reviewer_uXLc · 2023-07-26

**Soundness:** 3 good
**Presentation:** 3 good
**Contribution:** 3 good
**Rating:** 7
**Confidence:** 2

**Summary:**

This paper presents a method for end-to-end learning of metrics to train prediction models. It proposes a concept of task-driven metric learning. The main idea is to let the model focus on task-relevant features and dimensions in the prediction space. In addition, the resulting prediction model can be more interpretable, as the proposed method uses metric learning as a preconditioning step for gradient-based model training. Experiments have shown the effectiveness of the proposed method.

**Strengths:**

- The proposed idea of only modifying the metric that the prediction model uses for training is quite interesting. It enables the model to focus more on task-relevant features and dimensions without explicitly altering the optimal prediction model itself.  Hence, it allows the prediction model trained on its original prediction problem while being valuable to the downstream tasks.

- The results are promising. The proposed method has outperformed the other baseline approaches.

**Weaknesses:**

I'm not an expert in this field.
- The writing of the methodology is a bit difficult to understand. It would be great if the writing can be polished with more illustrations.

**Questions:**

I'm not an expert in this field.
The proposed method proposes to parametrize generalized Mahalanobis distance-based loss for training the prediction model.
Why do have to use generalized Mahalanobis distance-based loss?
Are there any other options for the metric? If yes, it would be better to provide some discussion and comparisons.

**Limitations:**

The paper has provided the limitations of the proposed method.

---

> ### Author Rebuttal · Authors · 2023-08-09
>
> Thanks very much for the review! We are delighted you liked the idea and found the results promising. Here are a few responses to your comments, please let us know if there is anything else:
>
> > The writing of the methodology is a bit difficult to understand. It would be great if the writing can be polished with more illustrations.
>
> We agree the methodology may be difficult to read without broader knowledge about other end-to-end DFL and implicit differentiation methods. We’ll make a pass through and try to explain as much as possible so it’s easier to understand, and we’ll think about places we can add illustrations. Are there any particular places in the methodology where you would want polishing?
>
> > The proposed method proposes to parametrize generalized Mahalanobis distance-based loss for training the prediction model. Why do have to use generalized Mahalanobis distance-based loss? Are there any other options for the metric? If yes, it would be better to provide some discussion and comparisons.
>
> This is a very good question that we will clarify in the paper. For us, the generalized Mahalanobis distance was a natural extension of the MSE loss: any parameterization of the Mahalanobis loss retains the optimal solution of the MSE loss and the metric learning only changes how modeling errors are weighted, which is learned using task gradients.
> Beyond the parameterized Mahalanobis distance, many other parameterized losses could be learned. [This paper](https://arxiv.org/abs/1906.05374) on meta-learning losses simply uses a neural network as the loss and uses the meta-learning signal to update it. While we didn’t go to this level of generality because the neural network loss loses some interpretability, exploring other loss models are certainly an interesting direction for future work, and perhaps there are other intermediates between the Mahalanobis and neural network losses.

---

> > ### Comment · Reviewer_uXLc · 2023-08-17
> > **Post rebuttal**
> >
> > Thanks for the rebuttal! The rebuttal has addressed my concerns -- I will keep my previous score.

---

### Author Rebuttal · Authors · 2023-08-10

We extend our gratitude to the reviewers for investing their time and expertise into our paper. The feedback provided has been immensely valuable, and we greatly appreciate the positive remarks regarding the novelty and impact of our proposed method. While we include more detailed responses inline below, we wish to highlight and emphasize the following points:

1. We are encouraged to see that the *majority of the reviewers resonated with us on the idea* – learning a metric end-to-end with a task to train models and also found the idea well corroborated with the experiments. We are delighted that our contributions and the soundness of our method have been recognized, as indicated by the favorable ratings of 3 (good) in both the areas.
2. *On the evaluation of our method in more challenging and real-world scenarios:* We would like to highlight that the decision focused model learning tasks are considered difficult and are widely used as benchmarks in task-based model learning literature. Since they incorporate solving combinatorial optimization problems, they represent a wide variety of real-world downstream optimization problems. Furthermore, while our model-based learning experiments may appear modest in scale, they effectively demonstrated the method's ability to distinguish between relevant and irrelevant features, as well as rank the task relevant features in accordance to their task relevance. At the same time, we would agree that it will be interesting to showcase these capabilities on more complex RL systems, which we assure to work towards.
3. Lastly, we wish to emphasize the distinct capability of TaskMet in case any reviewer overlooked – *achieving task performance on par with or exceeding that of DFL methods, while having minimal prediction error and making predictions in original prediction space unlike other DFL methods.* The idea – learning metric for the model loss function instead of learning model parameters using the task loss – allows TaskMet to achieve optimal task performance while being relevant for the original prediction problem.

In conclusion, we are genuinely appreciative of the reviewers' insights, which will be instrumental in refining our work. We look forward to more fruitful discussion in the coming week.

---

### Decision · Program_Chairs · 2023-09-21

**Decision:**

Accept (poster)

**Comment:**

This paper adapts the end-to-end task-based model learning paradigm by defining a metric loss that measures distance between model predictions and ground truth (similar to MSE minimization) but via a metric that is parameterized by task-based information. (This is in contrast to existing task-based learning approaches that weight between an MSE loss and a task-based loss.) Experimental results show that the approach achieves task performance competitive to task-based models, while still achieving prediction error (MSE error?) competitive to MSE-optimizing methods (i.e., without overfitting to the task). The reviewers agreed the work was interesting and well-motivated and the results were promising, and were unanimous in recommending acceptance. That said, they asked that the presentation be made more accessible (e.g., by polishing the writing and adding illustrations/flow charts/algorithm boxes) and recommended the implementation of larger-scale experiments to further validate the efficacy of the method.